# Detection of IDH1 Mutations in Plasma Using BEAMing Technology in Patients with Gliomas

**DOI:** 10.3390/cancers14122891

**Published:** 2022-06-11

**Authors:** Santiago Cabezas-Camarero, Vanesa García-Barberán, Rebeca Pérez-Alfayate, Isabel Casado-Fariñas, Hillary Sloane, Frederick S. Jones, Pedro Pérez-Segura

**Affiliations:** 1Head & Neck Cancer, Neuro-Oncology and Genetic Counseling Unit, Medical Oncology Department, Instituto de Investigación Sanitaria San Carlos (IdISSC), Hospital Clínico Universitario San Carlos, Paseo del Profesor Martín Lagos S/N, 28040 Madrid, Spain; pedro.perez@salud.madrid.org; 2Molecular Oncology Laboratory, Medical Oncology Department, Instituto de Investigación Sanitaria San Carlos (IdISSC), Hospital Clínico Universitario San Carlos, 28040 Madrid, Spain; vanesa.garciabar@salud.madrid.org; 3Department of Neurosurgery, Hospital Clínico Universitario San Carlos, 28040 Madrid, Spain; rebecap.alfayate@gmail.com; 4Pathology Department, Hospital Clínico Universitario San Carlos, 28040 Madrid, Spain; isabel.casadofa@salud.madrid.org; 5Medical affairs Division, Sysmex Inostics, Inc., Baltimore, MD 21205, USA; sloane.hillary@sysmex-inostics.com (H.S.); jones.fred@sysmex-inostics.com (F.S.J.)

**Keywords:** glioma, IDH1, liquid biopsy, circulating tumour DNA, ctDNA

## Abstract

**Simple Summary:**

In contrast with other solid tumors, only a few, small studies have shown the feasibility of detecting different biomarkers in the peripheral blood (PB) of patients with gliomas. A prospective study was conducted, enrolling 10 patients with gliomas where 33 consecutive PB samples were analyzed. Among the six patients with isocitrate dehydrogenase 1 (IDH1)-mutant tumors that were surveyed, circulating tumor DNA (ctDNA) was detected in PB in three of them (50%), at timepoints at which the patients were either untreated or exhibited progressive disease. While no false positives were identified, the false-negative rate was high, reaching 86% (18/21). Finally, in one of the IDH1-mutant cases, the Beads, Emulsion, Amplification and Magnetics (BEAMing) digital PCR technology detected one of the two IDH1 mutations that had been detected in the patient’s tumor sample in plasma, 7 years prior to its detection in blood.

**Abstract:**

Molecular testing using blood-based liquid biopsy approaches has not been widely investigated in patients with glioma. A prospective single-center study enrolled patients with gliomas ranging from grade II to IV. Peripheral blood (PB) was drawn at different timepoints for circulating tumour DNA (ctDNA) monitoring. Next-generation sequencing (NGS) was used for the study of isocitrate dehydrogenase 1 (IDH1) mutations in the primary tumor. Beads, Emulsion, Amplification and Magnetics (BEAMing) was used for the study of IDH1 mutations in plasma and correlated with the NGS results in the tumor. Between February 2017 and July 2018, ten patients were enrolled, six with IDH1-mutant and four with IDH1 wild-type gliomas. Among the six IDH-mutant gliomas, three had the same IDH1 mutation detected in plasma (50%), and the IDH1-positive ctDNA result was obtained in patients either at diagnosis (no treatment) or during progressive disease. While the false-negative rate reached 86% (18/21), 15 out of the 18 (83%) plasma-negative results were from PB collected from the six IDH-mutant patients at times at which there was no accompanying evidence of tumor progression, as assessed by MRI. There were no false-positive cases in plasma collected from patients with IDH1 wild-type tumors. BEAMing detected IDH1 mutations in the plasma of patients with gliomas, with a modest clinical sensitivity (true positivity rate) but with 100% clinical specificity, with complete agreement between the mutant loci detected in tumor and plasma. Larger prospective studies should be conducted to expand on these findings, and further explore the clearance of mutations in PB from IDH1-positive patients in response to therapy.

## 1. Introduction

Diffuse low- and high-grade gliomas constitute the 15th solid tumor in terms of incidence and the 10th in terms of mortality [1]. Despite the infrequent incidence of glioma in the population, the high mortality rate and dramatic neurologic sequelae make improvements in diagnostic and therapeutic options urgent. At present, the molecular classification of gliomas is mandatory to predict outcome and in therapeutic decision-making, as patients with isocitrate dehydrogenase (IDH)-mutant gliomas have a significantly better prognosis as compared to IDH-wild-type tumors, with a differentiated mutational profile between the two groups. This is reflected in the new WHO classification of central nervous system (CNS) tumors [2,3,4,5]. The IDH mutations are present in one third of patients with gliomas, with IDH1 and IDH2 mutations occurring in 70% and 10% of lower-grade gliomas, respectively [6]. Most IDH mutations are heterozygous missense mutations that promote the conversion of α-ketoglutarate to (R)-2-hydroxyglutarate ((R)-2HG), an oncometabolite with several biologic effects, such as cellular differentiation and chromatin methylation, via the inhibition of histone demethylases [5,7]. IDH mutations occur very early in the carcinogenic process and probably drive genetic instability and mutations in other oncogenes [5,7,8,9]. Indeed, Pappula et al. [5], described a clearly differentiated molecular profile between IDH-mutant and IDH-wild-type tumors using the COSMIC database. Interestingly, IDH mutations must be heterozygous to produce the (R)-2HG byproduct, and this is the reason for the better prognosis of IDH-mutant gliomas [5,7]. Several IDH inhibitors have been tested in other IDH-mutant malignancies and are currently being tested in patients with gliomas (NCT04056910, NCT03343197) [10,11,12,13]. Therefore, besides a therapeutic target, IDH mutations constitute ideal hotspot alterations, which can be tracked through liquid biomarker analyses [14,15,16]. Liquid biopsy allows for the spatial and temporal biases of traditional tissue biopsy to be circumvented, which is especially important in highly heterogenous diseases such as gliomas [5,16,17]. Due to the high morbidity associated with brain tumor biopsies, obtaining cerebrospinal fluid (CSF) for molecular analysis emerged as a less-invasive way to access the molecular profile of the tumor [16,17,18]. Indeed, the detection of ctDNA in CSF has been shown to precisely mirror tumor mutations from the primary tumor in patients with gliomas [16,19,20,21]. However, CSF draws are uncomfortable and potentially troublesome for patients, and require numerous health resources. Therefore, liquid biopsy in the peripheral blood is the ideal liquid biopsy modality for most cancer patients and especially for those with gliomas, given their commonly frail, dependent and neurologically deteriorated status [14,15,22,23,24,25,26,27,28,29,30,31]. However, liquid biopsy in gliomas needs to overcome several obstacles. In addition to lacking the morphological information provided by traditional tissue biopsy, detection methods need to gain sensitivity due to the blood–brain barrier (BBB) effect and the fact that gliomas commonly do not metastasize and have a lower tumor size than extra-CNS tumors, which limit ctDNA shedding to the peripheral blood [16,17]. To date, only a few studies have demonstrated that circulating tumor cells, extracellular vesicles and/or ctDNA can be detected in the peripheral blood of patients with gliomas [14,15,17,22,24,25,26,27,28,29,30,31]. However, these studies are hampered by their low sensitivity, with ctDNA detection rates usually below 15% [29,30]. Beads, Emulsion, Amplification and Magnetics (BEAMing) is a highly sensitive type of digital PCR, that emulsifies the PCR product, followed by the differential hybridization of mutant and wild-type DNA fragments with fluorescent magnetic beads, and before an analysis by fluid cytometry. This technology allows for the detection of 1 mutant among 10.000 wildtype alleles and is specifically designed for the detection of recurrent hotspot mutations, such as RAS, EGFR, PIK3CA, or IDH mutations in plasma, among different solid tumors and hematologic malignancies [32,33,34,35,36]. Given that BEAMing is currently the most sensitive technology for ctDNA detection in plasma, and due to the very limited evidence on liquid biopsy in primary brain tumors, we hypothesized that BEAMing could allow for the detection of IDH mutations in the peripheral blood of patients with gliomas [32]. To our knowledge, the present study is the first to use BEAMing technology for ctDNA detection in plasma in patients with low- and high-grade gliomas.

## 2. Materials and Methods

### 2.1. Patient Enrollment and Sampling

Only patients diagnosed with a newly diagnosed or untreated recurrent glioma from grade II to IV were enrolled in the study. No healthy controls were included. Patients with IDH1-wild-type tumors behaved as controls for the main purpose of this study: the identification of IDH1 mutations in plasma.

The fifth edition (2021) of the WHO Classification of Central Nervous System tumors was used for the histological and molecular classification of patients’ tumors [4].

Peripheral blood (PB) samples were prospectively collected at different timepoints of clinical interest in both newly diagnosed and recurrent disease (prior to surgical resection, before and after irradiation and before, during and after chemotherapy) (see the sample collection chronogram in Appendix A). PB samples were collected in 10-mL Streck cell-fre DNA BCT tubes (La Vista, NE, USA). PB samples and the corresponding formalin-fixed paraffin-embedded (FFPE) tumor samples were analyzed for IDH1 mutational status using BEAMing digital PCR and next-generation sequencing (NGS). This study was conducted according to the REporting recommendations for tumour MARKer prognostic studies (REMARK) [37].

### 2.2. Immunohistochemistry

IDH1-R132H antibody (clone H09) (Catalogue Reference: IDAH09, Dianova, Hamburg, Deutschland) was used to study IDH1-R132H mutation expression by immunohistochemistry (IHC) in an autostainer Dako Omnis (Agilent Technologies, Inc., Santa Clara, CA, USA), as previously described [38]. IDH1-R132H mutation expression was considered positive when tumor cells showed a strong cytoplasmic staining, while weakly diffuse tumour cell staining was considered negative [38].

### 2.3. DNA Extraction from Tissue and Plasma Samples

Tumor samples were obtained by open surgical resection and immediately processed by a pathologist. After macroscopic tumor selection, samples were fixed in formalin and paraffin-embedded (FFPE). The tumor region was selected and marked by a pathologist in an H&E section slide that was representative of the cellularity of samples collected from 4–8 FFPE sections that were 4–5 μm each. Tumour DNA was obtained from the areas marked by the pathologist from these 4–8 FFPE sections using the QIAamp DNA FFPE GeneRead Kit (Catalogue Reference: 180134; Qiagen, Germantown, MD, USA). DNA from plasma was purified using the QIAamp Circulating Nucleic Acid Kit (Catalogue Reference: 55114; Qiagen, Germantown, MD, USA). Tumor and plasma DNA were quantified using a QUBIT 3.0 fluorometer instrument and the Qubit 1× dsDNA HS Assay Kit (Catalogue Reference: Q33230; Thermo Fisher Scientific, Waltham, MA, USA).

### 2.4. Next Generation Sequencing (NGS) Study of IDH1 and IDH2 Mutations in Tumor Tissue

DNA from 4–8 sections (4–5 μm each) of formalin-fixed paraffin-embedded (FFPE) tumor samples (which were selected by a pathologist as described above) was extracted using the “QIAamp DNA FFPE GeneRead Kit” (Catalogue Reference: 180134; Qiagen, Germantown, MD, USA). Subsequently, the whole exonic region of IDH1 and IDH2 genes was then sequenced using an Illumina MiSeq device (version 3.1.0.13) following the manufacturer’s instructions, as previously described [39]. In brief, after DNA quantification from FFPE tumor samples using Qubit 1× dsDNA HS Assay Kit (Catalogue Reference: Q33230; Thermo Fisher Scientific, Waltham, MA, USA), 50–150 ng was used for mutational analysis by AmpliSeq methodology (Illumina, Inc., San Diego, CA, USA). AmpliSeq Library PLUS was used for library preparation (Catalogue Reference: 20019101; Illumina, Inc., USA), followed by the amplification of target regions and second amplification of libraries, which were diluted and denatured for bridge clonal amplification and paired-end sequencing using MiSeq Reagent kit v2 (300-cycles) (Catalogue Reference: MS-102-2002; Illumina, Inc., USA) in a MiSeq instrument (Illumina, Inc., USA). Variant calling files annotation, and the identification and classification of detected genetic variants were performed with the VariantStudio software v3.0 (Illumina, Inc., USA).

### 2.5. ctDNA BEAMing Digital PCR Analyses of IDH1 Mutations in Plasma

For variant allele frequency (VAF) assessment, the BEAMing dPCR assay (OncoBEAM; Sysmex Inostics Inc., Baltimore, MD, USA) was used to evaluate IDH1 mutations at position R132 (C/G/L/S/H), the most commonly mutated locus in gliomas [40]. The BEAMing technology has a lower limit of detection (LOD) for mutant IDH1 (mIDH1) alleles of 0.02% to 0.04% (2 × 10^−4^ to 4 × 10^−4^) VAF and is from 50- to 100-fold more sensitive than NGS [41]. BEAMing is a highly sensitive digital PCR method in which the PCR amplification is performed on beads in a water-in-oil emulsion. After amplification, emulsions are broken and either mutant or wild-type IDH1 DNA molecules are detected using specific fluorescently labeled hybridization probes, while attached to beads. The mutant fraction bound to the fluorescent beads is then analyzed by flow cytometry [32]. In brief, 150 μL of the PCR reaction was mixed with 600 μL of oil/emulsifier mix and added to a 96 deep-well plate 1.2 mL (Catalogue Reference: AB1127; Thermo Fisher Scientific, Waltham, MA, USA), followed by plate shaking, and then emulsions were dispensed in 8 PCR wells, followed by several PCR cycles. Emulsions were broken using a 150-μL breaking buffer added to each well. Beads were recovered after spinning the suspension and removing the oil phase. The DNA on the beads was denatured, followed by allele-specificic hybridization using fluorescently labeled probes targeting the mutant and wild-type DNA sequences designed for five different IDH1-R132H mutations. After incubating and cooling the hybridization mixture, the beads were separated with a magnet and finally resuspended in 200 μL of TE buffer to undergo flow cytometry analysis in a CyFlow^®^ Cube 6i cytometer (Sysmex Inostics, Inc., Baltimore, MD, USA) that separates mutant DNA-bound beads from those containing wild-type (unmutated) beads. The number of mutant and wild-type IDH1 beads were finally counted using the FCS Express software. The ratio of mutant/wild-type beads accurately represents the ratio of mutant/wild-type DNA obtained from the patient’s plasma sample [32].

### 2.6. Tumor Response Assessment

Brain magnetic resonance imaging (MRI) was performed in all patients as per standard of care and according to the response assessment in neuro-oncology (RANO) criteria [42,43,44]. Standard axial T1-weighted, T2-weighted FLAIR and contrast T1-weighted images were obtained at every MRI assessment and reviewed by an experienced neuroradiologist who was unaware of the plasma ctDNA results.

### 2.7. Ethical Considerations

This study was approved by the Institutional Review Board (IRB) of Hospital Clínico Universitario San Carlos (IRB code 16/549-E), in accordance with the principles outlined in the “World Medical Association Declaration of Helsinki”. A signed informed consent form was obtained from the subjects involved in this study prior to study participation.

## 3. Results

### 3.1. Baseline Characteristics of the Patients

Between February 2017 and July 2018, 30 patients were prospectively enrolled, of which 10 had at least ≥3 mL of plasma collected for ctDNA analysis using the OncoBEAM IDH1 assay and were followed until May 2022. Five patients were newly diagnosed, whereas five presented with recurrent disease. The malignancies observed in this cohort included 4 IDH-wild-type glioblastomas, 3 IDH-mutant astrocytomas (grade 4), 1 IDH-mutant astrocytoma (grade 3), 1 IDH-mutant astrocytoma (grade 2), and 1 IDH-mutant, 1p/19q-codeleted oligodendrogliomas (grade 2).

### 3.2. Immunohistochemistry Study of the IDH1-R132H Mutation in Tumor Tissue

The IDH1-R132H mutation was detected by immunohistochemistry (IHC) in 5 patients, with the remaining 5 patients being wild-type. Among 3 O6-methylguanine–DNA methyltransferase (MGMT)-evaluable glioblastomas, 2 were unmethylated and 1 was MGMT-methylated. These and other patient characteristics are summarized in Table 1.

### 3.3. NGS Study of IDH1 and IDH2 Mutations in Tumor Tissue

Using NGS, seven patients harbored heterozygous IDH1 mutations in tumor tissue, including 5 patients with R132H, 1 patient with R132G, and 1 patient with R132C. One patient harbored two co-occurring mutations in IDH1 (R132H and R132C). No mutations were detected in IDH2. Five patients had astrocytoma (one grade 2, one grade 3 and four grade 4) and 1 patient had a grade 2 oligodendroglioma (Table 1 and Table 2, Appendix A).

### 3.4. BEAMing Study of IDH1 Mutations in Plasma

Among the 10 patients, a total of 33 PB samples were obtained and analyzed, with a median of three serial PB samples per patient (Min-max: 1–7). Among the six IDH1-mutant patients identified by tumor tissue NGS, BEAMing detected the corresponding IDH1 mutation in the plasma of three patients at a single timepoint in each case. Two of these patients had grade 4 astrocytoma, one of whom was plasma ctDNA-positive after a partial tumor resection surgery but prior to starting concomitant first-line chemoradiation, and one patient who was plasma ctDNA-positive at the time of overt progression (leptomeningeal dissemination). The third patient was ctDNA-positive for the R132C mutation and had a recurrent grade II oligodendroglioma progressing after radiotherapy, where the primary tumor biopsy from 7 years earlier (the last surgery that had been performed) harbored two IDH1 mutations (R132H and R132C). While BEAMing detected the IDH1 mutation with lower VAF in the primary tumor, its value was above the LOD for the BEAMing technology (≥0.02%). Finally, 83% (15/18) of the plasma ctDNA-negative samples obtained from the six patients with IDH1-mutant tumors (NGS) occurred in patients with treated disease or without evidence of tumor progression (Table 1 and Appendix A). Figure 1, Figure 2 and Figure 3 depict the tumor evolution and timing of PB draws and IDH1 mutation detection in the three plasma-positive cases (Table 2). Figure 4 shows the BEAMing plots for each mutation detected in plasma in three ctDNA-positive patients. Figure 5 summarizes the main results of the study.

## 4. Discussion

Most studies that have successfully demonstrated clinical value for the use of liquid biopsy in brain tumors have used CSF as opposed to peripheral blood. Most of these studies showed a good correlation with tissue-based results, generally showing >60% concordance between the mutational results obtained from CSF with those obtained from the primary tumor [16,19,20,21]. In addition, some authors have demonstrated that changes in the CSF-based mutational profile accurately reflect the evolution of the primary tumor, which may be valuable in treatment monitoring and therapeutic decision-making [5,10]. However, the development of liquid biopsy in primary brain tumors has been limited by the molecular heterogeneity of the disease and the lack of recurrent hotspot mutations except for those in IDH, first described in 2009, as well as the limited ctDNA shedding due to the effect of the BBB and the CNS-confined nature of these disease [16,17]. Therefore, it is not surprising that only a few studies have been successful in detecting ctDNA in peripheral blood in patients with primary brain tumors, particularly gliomas [14,15,17,21,22,29,30,31]. Two pan-tumor NGS-based ctDNA studies using peripheral blood samples reported low rates of ctDNA detection in patients with gliomas, ranging from less than 10% to 15% [29,30]. On the other hand, a French group reported 60% concordance between the primary tumor and ctDNA assessed using COLD digital PCR for the detection of IDH1-R132H mutations. In addition, this study showed a higher ctDNA detection rate among high-grade gliomas and among high- vs. low-volume tumors. ctDNA methylation, a different ctDNA liquid biopsy modality, has also been studied [14]. Lavon et al. [31], demonstrated a moderate sensitivity and specificity in the identification of loss-of-heterozygosity (LOH) and MGMT/PTEN methylation in peripheral blood in patients with gliomas, detecting MGMT methylation in up to 24% of patients. However, 24% of the patients were methylation-positive in serum, despite showing no evidence of tumor via MRI. Interestingly, the ctDNA methylome was recently explored in patients with intracranial tumors, demonstrating a high accuracy to distinguish among different intra- and extra-cranial tumors and between IDH-wild-type and IDH-mutant gliomas [22]. Although ctDNA methylome analysis seems promising as a diagnostic tool in patients with intracranial tumors, to our knowledge, this methodology is not routinely implemented, limiting its current application. In contrast to the data presented in our study, none of the studies referenced above have performed serial ctDNA analysis, which is likely to be of significant clinical value in longitudinal monitoring of the disease.

Other tumor components have been detected in the peripheral blood in patients with gliomas [17]. Two studies demonstrated the shedding of Circulating Tumor Cells (CTCs), characterized by stem- and mesenchymal-like features in patients with glioblastoma. However, the relatively low number of CTCs, combined with technical difficulties in their detection, limit the utility of this approach [27,28]. Another research group demonstrated that exosomes from a low-grade glioma murine model were able to cross the blood–brain barrier (BBB). In patients with IDH-mutant gliomas, investigators could detect tumor exosomes in the peripheral blood and analyze their cargo to successfully detect the IDH1-R132H mutation with a high concordance rate [15] (Table 3).

In our study, among six IDH1-mutant patients identified by tissue NGS, BEAMing detected the same mutation in the plasma of three of these patients (50%), reaching a specificity of 100%. The true-positive rate achieved 14.3% (3/21) among the 21 PB draws; therefore, the false-negative rate was high (86.4%). However, it must be noted that in 15 out of the 18 ctDNA-negative plasma samples from the 6 IDH1-mutant patients, PB was collected at timepoints without evidence of progressive disease on MRI. Therefore, it could be speculated that if PB was collected only with untreated or progressive disease, the false-negative rate would be lower. Although the mutations were detected in ctDNA at a single timepoint in each of the patients, all three ctDNA-positive patients had either untreated or progressive disease (3/6: 50%), indicating BEAMing’s value in detecting plasma ctDNA in cases of active disease. In the two ctDNA-positive IDH1-mutant grade 4 astrocytoma patients, IDH1 mutations were not detected at other timepoints, but no tumor progression was observed in MRI for any patient. In the ctDNA-positive patient with an oligodendroglioma, the two IDH1 mutations were not detected pre-radiotherapy, nor were they detected in the second post-radiotherapy timepoint while viable disease was present in both cases. However, this patient had a lower-grade tumor and a much lower disease burden than the other two patients. Moreover, lower-grade gliomas are known for being slow-progressing tumors and thus, possibly, ctDNA low-shedding tumors [2,3,45]. In addition, lower-grade gliomas are known for experiencing delayed responses to both radiation and chemotherapy; this could also explain the lack of ctDNA detection in the second post-RT timepoint (Figure 3) [45]. Interestingly, this patient, with a 15-year history of a grade II oligodendroglioma, that underwent surgery in 2004 and 2011 and eventually relapsed in 2017, after being subjected to radiotherapy was shown to harbor two IDH1 mutations (R132H, R132C) in the 2011 biopsy, one of which—the one with the lowest VAF in tissue—was detected in plasma in 2017. This demonstrates the liquid biopsy’s power to reveal the molecular heterogeneity and evolution of the tumor [16,21]. Indeed, while the R132H mutation had a high VAF in both tumor and plasma, the other mutation (R132C) had a much lower, but still detectable, VAF—above the NGS and BEAMing LOD—in both tumor and plasma. Although our patient had not been treated with any IDH inhibitor, this finding is consistent with the observation reported by other investigators, describing the emergence of other IDH mutations as a resistance mechanism in IDH-mutant leukemias treated with selective IDH-inhibitors; this either indicates newly developed resistance mutations or the expansion of co-existing but sub-clonal IDH mutations [46,47]. Since we only studied tumor somatic alterations and did not perform germline DNA studies or conduct single-cell analysis, we are unable to demonstrate if the two co-occurring mutations in IDH1 in this patient belonged to a homozygous clone or to two different clones. However, it is very unlikely that this patient harbored a homozygous IDH1-mutant cell clone, given the very different VAFs for each mutation in the primary tumor, and that only one mutation (R132C) was detected in plasma. If this tumor cell clone had been homozygous, with two different IDH1-mutant alleles (R132H and R132C), both mutations would have been detected in tumor and plasma with a similar VAF. In addition, IDH-mutant tumors need to be heterozygous in order to promote the conversion of α-ketoglutarate into the oncometabolite (R)-2HG) [5,7]. Therefore, in our patient, it is more probable that the R132H mutation belonged to a predominant IDH1-mutant clone and the R132C corresponded to a sub-clonal cell population. To our knowledge, our study is the first report showing two co-existing IDH1 mutations detected in tissue in a patient with a glioma, and it is also the first to detect the delayed emergence of an IDH1 mutation in plasma that was sub-clonally detected in tissue 7 years earlier. Considering that IDH co-occurring mutations in different IDH1 loci or in IDH1 and IDH2 loci are extremely rare events in gliomas, our finding is of special interest to the field. Indeed, Hartmann et al. [6], in a study of 747 IDH-mutant gliomas, found only four patients with two co-occurring IDH1 and IDH2 mutations, while there were no cases of two or more co-occurring IDH1 mutations.

Our study is limited by its small sample size and the low number of blood draws performed in some patients, which possibly could have detected IDH1 mutations in some of them. In addition, we could not study ctDNA mutations in the IDH2 gene, which accounts for nearly 10–15% of the total of IDH mutations in gliomas, although none of the 10 patients harbored IDH2 mutations in the primary tumor [48]. A majority (83%) of the ctDNA-negative samples performed in the NGS-IDH1-mutant patients were obtained at timepoints at which the disease was either stable or responding, possibly explaining the absence of IDH1-mutant ctDNA. Since different methods were used for the analysis of IDH mutations in tumor tissue and in plasma in this study, it would have been interesting to compare NGS and BEAMing performance for the detection of IDH mutations in the primary tumor. As in studies that were previously performed in IDH-mutant leukemias [10,41], future studies should investigate whether the highly sensitive BEAMing methodology may be able to detect co-occurring clonal or sub-clonal IDH mutations in tumor tissue, as well as co-ocurring mutations in other genes that elude detection by NGS but may aid in the more accurate mutational profiling of these tumors [5,40,41]. Since IDH mutations occur very early in the carcinogenic process and mutant IDH probably behaves as a driver gene, promoting genetic instability and mutations in other oncogenes, and thereby establishing a clearly differentiated molecular profile in IDH-mutant compared to IDH-wildtype tumors, it would have been of interest to study other co-occurring mutations in plasma in order to increase the diagnostic accuracy of liquid biopsy in our study [2,3,5,6,7,8,9,16]. Indeed, expanding the number of genes that are studied might also increase the likelihood of detecting tumor mutations in a larger number of patients with gliomas; this might enable a more refined approach to disease-monitoring and tumor heterogeneity evaluation [16]. Finally, we were unable to obtain CSF samples at the same time as the blood samples in order to study the correlations between the tumor, the CSF and the plasma [21].

## 5. Conclusions

This study demonstrates the feasibility of BEAMing technology to detect plasma IDH1 mutations in patients with IDH1-mutant gliomas for the first time. Detection in plasma occurred in the presence of either untreated or progressive disease, with no false-positive cases being identified. BEAMing could serve as a powerful liquid biopsy technology for ctDNA detection in plasma and CSF in patients with gliomas and should be evaluated in a larger, prospective study.

## Figures and Tables

**Figure 1 cancers-14-02891-f001:**
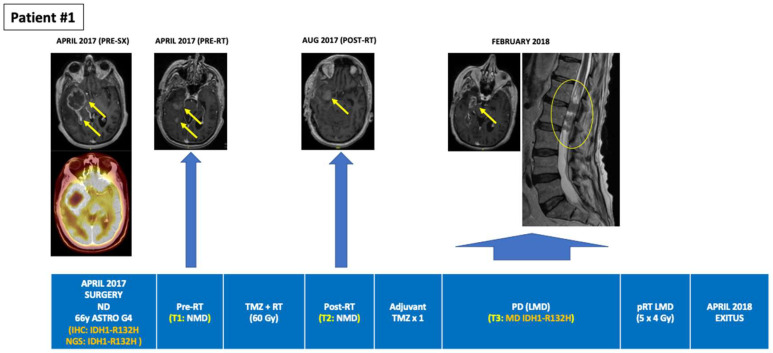
Tumour evolution in BEAMing ctDNA-positive patient #1. A 35-year-old man was diagnosed with a large left temporal IDH1-R132H mutant and MGMT unmethylated glioblastoma involving the Wernicke area. Partial tumor resection was followed by concurrent RT + TMZ then followed by 10 cycles of adjuvant TMZ. Prior to the start of RT + TMZ, peripheral blood was drawn for BEAMing ctDNA analysis, detecting a mutation in IDH1-R132H (VAF 0.071%). PB draws along 6 consecutive timepoints during adjuvant TMZ was negative for IDH1-mutations coincident with a sustained partial response on MRI. After adjuvant TMZ, a wait and see period was started with tumor progression occurring in November 2020. After two cycles of TMZ the tumor progressed, and the patient was subjected to a partial resection followed by hypofractionated RT (30 Gy in 10 fractions) between March and April 2021. In September 2021 treatment with bevacizumab was started achieving a partial response after two cycles (treatment currently ongoing). BEAMing: Beads, Emulsion, Amplification and Magnetics, ctDNA: circulating tumor DNA, IHC: immunohistochemistry, LMD: leptomeningeal disease, LOD: limit of detection, MD: mutation detected, MGMT: O6-methylguanine–DNA methyltransferase, NGS: next-generation sequencing, NMD: no mutation detected, pRT: palliative radiotherapy, RT: radiotherapy, SX: surgery, TMZ: temozolomide, Unmet: unmethylated, VAF: variant allele frequency.

**Figure 2 cancers-14-02891-f002:**
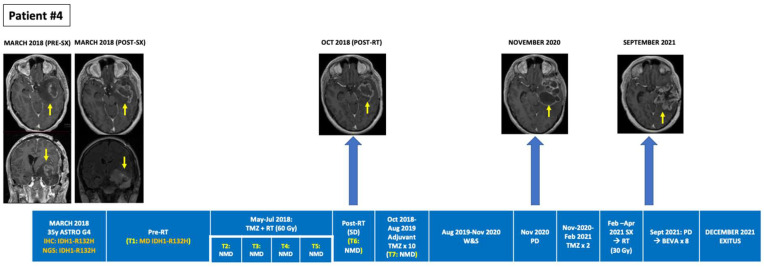
Tumour evolution in BEAMing ctDNA-positive patient #4. A 66-year-old man diagnosed with a large right temporal IDH1-R132H mutant and MGMT methylated glioblastoma underwent a subtotal resection. After surgery concomitant RT + TMZ was administered. A single adjuvant TMZ cycle was administered after RT due to clinical deterioration, partly due to a non-obstructive hydrocephalus that required a ventriculo-peritoneal valve placement. Plasma ctDNA analysis pre- and post-RT revealed no IDH1 mutations. Eight months after RT, in February 2018, progressive disease occurred with appearance of L1-L2 leptomeningeal dissemination that was treated with palliative RT (5 × 4 Gy). BEAMing plasma ctDNA analysis prior to L1-L2 irradiation, detected the IDH1-R132H mutation (VAF 0.377%). The patient died in April 2018, 12 months after diagnosis. BEAMing: Beads, Emulsion, Amplification and Magnetics, BEVA: bevacizumab, ctDNA: circulating tumor DNA, GBM: glioblastoma, IDH1: isocitrate dehydrogenase type 1, IHC: immunohistochemistry, MGMT: O6-methylguanine–DNA methyltransferase, MUT: mutant, NGS: next-generation sequencing, NMD: no mutation detected, RT: radiotherapy, SX: surgery, TMZ: temozolomide, VAF: variant allele frequency, W&S: wait and see.

**Figure 3 cancers-14-02891-f003:**
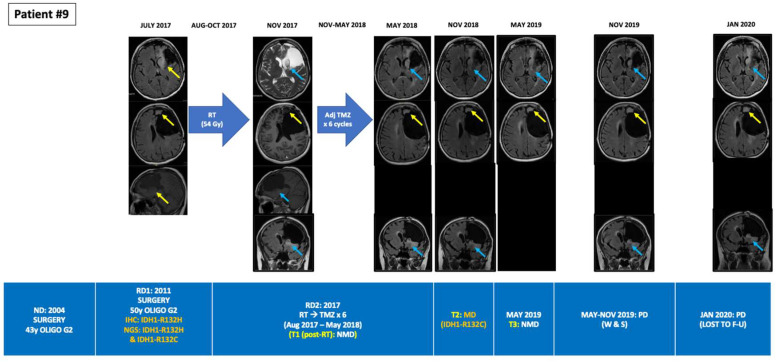
Tumour evolution in BEAMing ctDNA-positive patient #9. A 58-year-old woman was diagnosed in 2004 of a grade II oligodendrogliomas undergoing a gross total resection. In 2011 the tumor relapsed, undergoing a subtotal resection with aphasia as a postsurgical sequel. The patient then received adjuvant TMZ for 12 cycles. RT was disregarded due to the potential sequelae given the large volume to irradiate. In July 2017, the patient suffered an unresectable relapse. NGS analysis of the 2011 archived tumor sample unveiled two co-existing mutations in IDH1 (R132H and R132C). RT (54 Gy in 30 fractions) was administered between August and October 2017. BEAMing plasma ctDNA analysis did not detect any mutation prior to RT. After RT the patient received adjuvant TMZ for 6 cycles. Partial response was observed after RT, but progressive disease occurred in the antero-medial surgical cavity margin from May 2018 until January 2020, when the patient was lost to follow up. In May 2018, after finishing adjuvant TMZ, BEAMing plasma ctDNA analysis detected one of the IDH1 mutations detected in the tumor resected in 2011 (ctDNA IDH1-R132C VAF 0.025%, which is above de LOD for BEAMing (>0.02%)). However, ctDNA analysis performed in November 2018 did not detect any mutation. BEAMing: Beads, Emulsion, Amplification and Magnetics, ctDNA: circulating tumor DNA, F-U: follow-up, IDH1: isocitrate dehydrogenase type 1, IHC: immunohistochemistry, LMD: leptomeningeal disease, LOD: limit of detection, MUT: mutant, NGS: next-generation sequencing, NMD: no mutation detected, pRT: palliative radiotherapy, RT: radiotherapy, SX: surgery, TMZ: temozolomide, Unmet: unmethylated, VAF: variant allele frequency.

**Figure 4 cancers-14-02891-f004:**
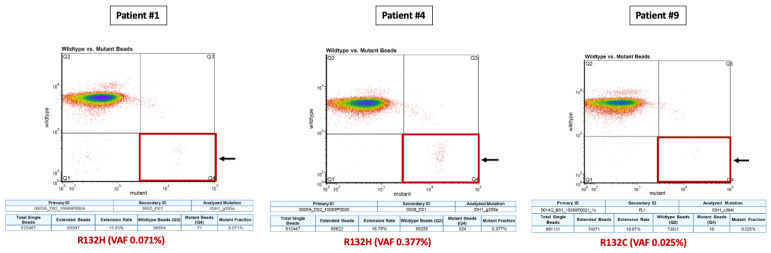
BEAMing ctDNA plots showing the IDH1 mutations detected and their respective VAF Red squares and black arrows indicate the mutant fraction within the BEAMing plot. BEAMing: Beads, Emulsion, Amplification and Magnetics, IDH1: isocitrate dehydrogenase type 1, VAF: variant allele frequency.

**Figure 5 cancers-14-02891-f005:**
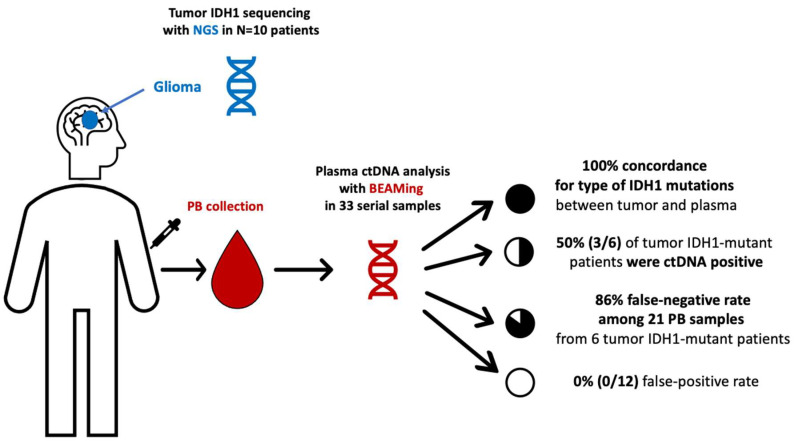
Summary of the results of the mutational study of IDH1 using NGS in tumor and using BEAMing in plasma in patients with gliomas. BEAMing: Beads, Emulsion, Amplification and Magnetics, IDH1: isocitrate dehydrogenase type 1, PB: peripheral blood.

**Table 1 cancers-14-02891-t001:** Summary of patient characteristics at the time of IDH1 mutational study in plasma.

Variable			Value
N			10
Male:Female			6:4
Median Age (range)	At initial diagnosis		51 (31–78)
	At blood collection		58 (34–78)
Histology ^†^	Glioblastoma, IDH-wild-type		4
	Astrocytoma (grade 4), IDH-mutant		3
	Astrocytoma (grade 3), IDH-mutant		1
	Astrocytoma (grade 2), IDH-mutant		1
	Oligodendroglioma (grade 2), IDH-mutant, and 1p/19q-codeleted		1
IDH R132H (IHC)	MUT		4
	WT		6
IDH1 & IDH2 status (NGS)	MUT		6 (IDH1-R132H: 5; IDH1-R132G: 1; IDH1-R132C: 1 ^‡^)
	WT		4
MGMT (GB, IDH-WT)	Methylated		1
	Unmethylated		2
	Unknown		1
Treatment received prior to or during blood collection	Surgery	GTR	3
		Subtotal Resection	2
		Partial Resection	4
		Biopsy	2
	Radiotherapy	RT alone	2
		RT + TMZ	7
	Adjuvant TMZ (median No. of cycles)		8
Lobe/Area	Frontal		5
	Temporal		5
	Parietal/Occipital		0
	Cerebellar		1
Side	Left		7
	Right		3
	Bilateral		1
Scenario	Newly diagnosed		6
	Recurrent		5
Tumor NGS and BEAMing ctDNA analyses *	No. of peripheral blood draws	Total No. of PB draws	33
		Median PB per patient (range)	3 (1–7)
		Tumor NGS (+)	21/33
		Tumor NGS (−)	12/33
		NGS (+) and Untreated/PD disease	3/18 (17%)
		NGS (+) Treated/non-PD disease	15/18 (83%)
	Pts with concordance NGS (+) and ctDNA		3/6 (50%)
	Pts with concordance NGS (−) and ctDNA		4/4 (100%)
	ctDNA (+) rate in NGS (+) pts (true-positive rate)		3/21 (14.3%)
	ctDNA (−) rate in NGS (+) pts (false-negative rate)		18/21 (86.4%)
	Rate of NGS (+) ctDNA (+) and Untreated/PD		3/6 (50%)
	Rate of NGS (+) ctDNA (−) and Treated/non-PD		15/15 (100%)

Beads, Emulsion, Amplification and Magnetics (BEAMing), ctDNA: circulating tumor DNA, GBM: glioblastoma multiforme, GTR: gross total resection, IDH: isocitrate dehydrogenase, IHC: immunohistochemistry, VAF: variant allele frequency, MGMT: O6-methylguanine–DNA methyltransferase, MUT: mutant, PD: progressive disease, RT: radiotherapy, TMZ: temozolomide, WT: wild-type. ^†^ According to the 5th edition of the WHO classification of CNS tumors. ^‡^ Patient 10 carried two different IDH1 mutations in tumor (R132H and R132C). * Refers to IDH1/2 status in tumor analyzed by NGS and to IDH1 status in ctDNA analyzed with the OncoBEAM assay.

**Table 2 cancers-14-02891-t002:** Summary of IDH status in tumor and plasma for each patient.

Patient No.	Histology (Grade) ^†^	Age/Current Disease Setting/Treatment History	Time From IDx to ctDNA (Months)	Tumor IHC IDH1-R132H	Tumor NGS IDH1 (VAF)	Plasma ctDNA BEAMing IDH1 (VAF)(RANO)						
						T1	T2	T3	T4	T5	T6	T7
**1**	**Astrocytoma (grade 4), IDH-mutant**	65/ND/SX→RT + TMZ→TMZ × 2→PD (Feb 2018) →Exitus 12 m postND	**2**	**MUT**	** R132H ** (13.3%)	-	NMD (PR; Post-SX)	NMD (SD; Post-RT)	-	-	**R132H** (0.071%) (**PD; 10-m post-SX**)	-
2	Glioblastoma, IDH-WT	48/ND/SX→RT + TMZ→TMZ × 6→W&S→PD→SX→Exitus 12 m postND	2	WT	WT	NMD (SD; 2 m Post-RT)	NMD (SD; 4 m Post-RT)	-	NMD (PD; 7 m post-SX)	NMD (PD; 9 m post-SX)	NMD (PD; 10 m post-SX)	NMD (PD; 12-m post-SX)
3	Astrocytoma (grade 3), IDH-mutant	70/RD/SX (Jun 2015)→SX (2018)→TMZ × 12→W&S (Jan 2020)→SD (ongoing)	36	MUT	** R132H ** (14.35%)	NMD (**PD;** **Pre-BX**)	NMD (SD; 1 m Post-BX)	-	NMD (SD; Post 2nd TMZ)	NMD (SD; Post 4th TMZ)	NMD (SD; Post 5th TMZ)	NMD (SD; Post 6th TMZ)
4	Astrocytoma (grade 4), IDH-mutant	34/ND/SX (PR May 2018)→RT + TMZ→TMZ × 10→W&S (12 m)→TMZ × 2→SX (2021)→RT→PD (Oct 2021)→BEV (ongoing)	1	MUT	** R132H ** (31.1%)	**R132H** (0.377%) (**ND;** **Pre-RT**)	NMD (Intra-RT)	NMD (Intra-RT)	NMD (1 m Post-RT)	NMD (2 m Post-RT)	NMD (PsPD; Post 2nd TMZ)	NMD (Post 3rd TMZ)
5	Glioblastoma, IDH-WT	50/ND/SX→Exitus 2 m post-SX	0	WT	WT	NMD (ND; Pre-BX)	-	-	-	-	-	-
6	Glioblastoma, IDH-WT	55/RD/SX (2013)→RT + TMZ→TMZ × 6→SX (RD; 2017)→TMZ × 6→W&S (6 m)→BEV × 6 m→Exitus 6y postND	48	WT	WT	NMD (RD; Pre-SX)	NMD (RD; Pre-SX)	-	-	-	-	-
7	Glioblastoma, IDH-WT	72/ND/SX (2017)→RT + TMZ→TMZ × 3→SD→W&S (SD ongoing)	0	WT	WT	NMD (PD; Pre-SX)	NMD (PD; Pre-RT)	NMD (PsPD; 1 m Post-RT)	-	-	-	-
8	Astrocytoma (grade 2), IDH-mutant	34/RD/SX (2015)→RT + TMZ→TMZ ×→SX (RD; 2017)→BEV × 30→PD→Exitus 38 m postND	26	WT	** R132G ** (20.5%)	NMD (PR; Post-SX)	-	-	-	-	-	-
9	Oligodendroglioma (grade 2), IDH-mutant, and 1p/19q-codeleted	58/RD2/SX (ND; 2004)→SX (RD1; 2011)→RT (RD2; 2017)→TMZ × 6→W&S→PD & lost to FU in Jan 2020	156	MUT	** R132H ** (44.8%) **R132C** (0.75%)	NMD (**PD;** **Pre-RT**)	**R132C** (0.025%) (**SD *; 5 m Post-RT**)	NMD (**SD *; 11 m Post-RT**)	-	-	-	-
10	Astrocytoma (grade 4), IDH-mutant	78/ND/SX (2018)→RT + TMZ→TMZ × 3→PD→Exitus 7-m postSX	0	WT	** R132H ** (5.9%)	NMD (**ND;** **Pre-BX**)	-	-	-	-	-	-

Astro: astrocytoma, BEAMing: beads, emulsion, amplification and magnetics, BX: biopsy, ctDNA: circulating tumor DNA, GBM: glioblastoma multiforme, IDx: initial diagnosis, IHC: immunohistochemistry, MUT: mutant, ND: new diagnosis, NMD: no mutation detected, NGS: next generation sequencing, Oligo: oligodendroglioma, PD: progressive disease, PR: partial response, RANO: response assessment in neuro-oncology criteria, RD: recurrent disease, RD1: first RD, RD2: second RD, RT: radiotherapy, SX: surgery, T: time at which blood was drawn for ctDNA analysis, VAF: variant allele frequency, W&S: wait and see strategy, WT: wild-type. **Blue** indicates mutations in tumor tissue. **Red** indicates mutations in ctDNA. **Black bold** indicates tumor-NGS IDH1+ patients with ND or showing PD on MRI. (-) indicates either not performed or not applicable. ^†^ According to the 5th edition of the WHO classification of tumors of the CNS. * Patient with a paradoxical response to RT, with stable disease of the irradiated lesion and PD at the anteromedial surgical margin.

**Table 3 cancers-14-02891-t003:** Summary of studies of ctDNA in plasma in patients with gliomas.

Author (Year)	N	ctDNA Reservoir	ctDNA Detection Method	% of ctDNA-Positive Patients	% of ctDNA Positive Controls	Type of ctDNA Mutations Detected	Main Findings
Lavon (2010) [31]	Astro: N = 41 Oligo: N = 34	Plasma	10q LOH MGMT and PTEN methylation	Astrocytomas: -10q LOH: 51% -MGMTmet: 24% -PTENmet: 0% Oligodendrogliomas: -10q LOH: 79% -1p LOH: 17% -19q LOH: 4% -MGMTmet: 24%	24% serum positive with no viable tumor on MRI	-	Moderate sensitivity and specificity for LGG and HGG
Boisselier (2012) [14]	Gliomas: N = 39 (n = 25 IDH1m) HC: N = 14	Plasma	COLD dPCR (IDH1-R132H)	Total: 60% (15/25) LGG: 37.5% (3/8) HGG: 70.6% (12/17)	0 HC (0/14)	IDH1-R132H	Higher detection rate among HGG vs. LGG (70.6% vs. 37.5%) Higher detection rate and higher DNA concentration among high vs. low volume tumors
Bettegowda (2014) [30]	Intra- and extracranial cancers: N = 177 Gliomas: N = 27	Plasma	NGS	<10%	-	-	Extracranial malignancies: ctDNA detected in 82% Intracranial malignancies: ctDNA detected in <50% of MB and in >10% of gliomas
Schwaederle (2017) [29]	Intra- and extra-cranial cancers: N = 670 Gliomas: N = 152	Plasma	NGS	15% (pts with characterized actionable alterations)	-	-	Extra- and intra-cranial malignancies: ctDNA detected in 48% Gliomas: ctDNA detected in 15%
Nassiri (2021) [22]	IDHm gliomas: N = 70 IDHwt gliomas: N = 52 Menin- giomas: N = 60 Hemangio- pericytomas: N = 9 Low-grade glial-neuronal tumors: N = 14 Brain mets of UK primary: N = 9	Plasma	cfMeDIP-seq	-	-	-	High sensitivity and discriminative capacity between: Gliomas vs. other cancers vs. HC IDHm vs. IDHwt HGG vs. LGG High correlation between plasma and tumor methylation signatures
Current study (2022)	N = 10 (6 IDH1m, 4 IDH1wt)	Plasma	BEAMing for IDH1m	50% (3/6)	0 IDH1wt (0/4)	IDH1-R132H (n = 2) IDH1-R132C (n = 1)	Same mutant loci detected in IDH1 plasma (BEAMing) and tumor (NGS) All ctDNA+ pts had active disease on MRI In 1 pt BEAMing detected in plasma 1 of the 2 co-existing IDH1 mutations in tissue (R132H, R132C).

Astro: astrocytoma, BEAMing: Beads, Emulsion, Amplification and Magnetics, cfMeDIP-seq: cell-free methylated DNA immunoprecipitation and high-throughput sequencing, ctDNA: circulating tumor DNA, dPCR: digital PCR, HC: healthy controls, HGG: high-grade gliomas, IDH: isocitrate dehydrogenase, IDHm: IDH mutant, IDHwt: IDH wild-type, LGG: low-grade gliomas, LOH: loss of heterozygosity, mets: metastases, MGMT: O6-methylguanine–DNA methyltransferase, NGS: next-generation sequencing, oligo: oligodendroglioma, pt: patient, pts: patients, UK: unknown. (-): indicates either not reported or not applicable.

## Data Availability

Not applicable.

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
