# Peer review of "Detection of IDH1 Mutations in Plasma Using BEAMing Technology in Patients with Gliomas"

_cancers, 2022, doi:10.3390/cancers14122891_

Round 1
Reviewer 1 Report
To editors and authors
Detection of IDH1 mutations in plasma using beaming technology in patients with gliomas
This is a very interesting manuscript that should be considered for publication in CANCERS after some revisions below.
1) Please recheck and revise cautiously citation and references as MDPI format.
2) In each imaging, please add icon like arrow or arrowhead to point out the lesions.
3) Figure 1 is very small. Please enhance it.
4) I believe that imaging part lacked a very updated reference related to this manuscript strictly. PMID: 32588986
5) References in Table 3 need to be quoted in [number] not (number).
Sincerely
Author Response
REVIEWER 1:
To editors and authors
Detection of IDH1 mutations in plasma using beaming technology in patients with gliomas
This is a very interesting manuscript that should be considered for publication in CANCERS after some revisions below. ïƒ We thank the reviewer for his kind words and for the time dedicated to reviewing our manuscript which have allowed to improve and enrich our manuscript. Please see our responses below:
1) Please recheck and revise cautiously citation and references as MDPI format. ïƒ AUTHOR RESPONSE: Thank you for the suggestion. We have now corrected the citation and references by the MDPI format.
2) In each imaging, please add icon like arrow or arrowhead to point out the lesions. ïƒ AUTHOR RESPONSE: Thank you for the suggestion. We have now added an icon-like arrow to point out the lesions in each case.
3) Figure 1 is very small. Please enhance it. ïƒ AUTHOR RESPONSE: Thank you for the suggestion. We have divided Figure 1 into 4 different figures. We have separated the MRI images for each patient into an individual figure in order to magnify it and make it much easier to visualize. We have added a fourth figure including only the BEAMing figures from the three patients, making it much easier to visualize.
4) I believe that imaging part lacked a very updated reference related to this manuscript strictly ïƒ AUTHOR RESPONSE: Thank you for the suggestion. We have now added one more reference regarding this, as the author suggests.
5) References in Table 3 need to be quoted in [number] not (number). ïƒ AUTHOR RESPONSE: Thank you for the suggestion. We have now corrected this.
Sincerely

Reviewer 2 Report
Detection of IDH1 mutations in plasma using beaming technology in patients with gliomas (cancers-1739879)
In this manuscript, the authors basically follow 10 patients with gliomas (5 different types/grades) and access the pertinence of IDH1/2 gene mutations as registered by BEAMing / NGS.
This paper presents, very succinctly, somewhat well-characterised cases of IDH expression in gliomas (& respective methylation) and clinicopathological data.
Please, find in attachment some comments/suggestions/questions or curiosities of mine.
I do recognize that the small sample is also due to the difficulty of enrolling the "right" patients. Although this remains a downfall due to the very different biology of the tumours encountered. Nonetheless, the observations here reported are very meaningful. Controls used/consulted? Patients' own blood? (this should be clear).
A final illustration summarizing and depicting the authors' conclusions/theory is suggested.

Author Response
REVIEWER 2:
In this manuscript, the authors basically follow 10 patients with gliomas (5 different types/grades) and access the pertinence of IDH1/2 gene mutations as registered by BEAMing / NGS. This paper presents, very succinctly, somewhat well-characterised cases of IDH expression in gliomas (& respective methylation) and clinicopathological data.
I do recognize that the small sample is also due to the difficulty of enrolling the "right" patients. Although this remains a downfall due to the very different biology of the tumours encountered. Nonetheless, the observations here reported are very meaningful.
Please, find in attachment some comments/suggestions/questions or curiosities of mine.
AUTHOR RESPONSE: We thank the reviewer for the time dedicated to reviewing our manuscript and greatly appreciate all the suggestions that have allowed to improve and enrich our manuscript. Please see our responses below:
1) Controls used/consulted? Patients' own blood? (this should be clear) ïƒ AUTHOR RESPONSE: Thank you for the suggestion. The were no healthy controls in our study. The 4 tumor-IDH wildtype patients behaved as controls in our study. We have clarified it in lines 162-165.
2) A final illustration summarizing and depicting the authors' conclusions/theory is suggested. ïƒ AUTHOR RESPONSE: Thank you for the suggestion. We have added Figure 5 summarizing the results/conclusions of the study.
3) Simple summary:
-Line 19: “odetecting” ïƒ AUTHOR RESPONSE: Thank you for the suggestion. Typo corrected: “detecting”
-Line 23: 86% (28/21) ïƒ AUTHOR RESPONSE: Thank you for the suggestion. Typo corrected: 86% (18/21)
4) Keywords: The wording is a challenge. It should be reworded for clarity ïƒ AUTHOR RESPONSE: Thank you for the suggestion. Most of the Keywords have now been substituted by other more easy-to-find keywords.
5) Introduction:
-Line 48: …the molecular classification of gliomas is mandatory for the proper “prognostication” and therapeutic decision-making… ïƒ AUTHOR RESPONSE: Thank you for the suggestion. We have rewritten the sentence as: …the molecular classification of gliomas is mandatory for outcome prediction and therapeutic decision-making…
-Liquid biopsy in brain tumors also have cons! ïƒ AUTHOR RESPONSE: This is an important issue. Thank you for the suggestion. We have added a comment on this in Lines 84-89.
-Although flow cytometry is common, BEAMing technology is not widely applied. Therefore, a quick explanation on how it works and where it has shown proves would improve introduction ïƒ AUTHOR RESPONSE: This is an important issue. Thank you for the suggestion. We have added a brief paragraph explaining BEAMing in Lines 93-104.
-Plus, the importance of IDH1 mutations in brain cancers / gliomas (or other relevant tumor; since many primary tumours metastasize to the CNS/brain) ïƒ AUTHOR RESPONSE: This is an important issue. Thank you for the suggestion. We have added a paragraph on the importance of IDH mutations in gliomas in Lines 61-75.
6) Materials and Methods:
-Line 122: what to you mean by retrospectively in this context, specifically? ïƒ AUTHOR RESPONSE: Thank you for the suggestion. It was an error. It was a prospective study where patients were enrolled and samples were collected prospectively. Therefore, we have eliminated the word “retrospectively”.
- Section 2.2. Immunohistochemistry: cat ref# of IDH1-R132H antibody? ïƒ AUTHOR RESPONSE: Thank you for the suggestion. We have now added the cat ref#.
-Section 2.2. Immunohistochemistry: I have not found in this Ref any information on concentrations, dilutions, blocking, etc. I ain’t sure how you managed to reproduce this protocol? Also, clones are different (ref. 23 cites clone DIA-H09 from Dianova), thus information on controls would be helpful! ïƒ AUTHOR RESPONSE: Thank you for the suggestion. We have substituted the prior reference by a more appropriate one where all these methods are described in detail.
-Section 2.3. DNA extraction from tissue and plasma samples: What was the criteria of the pathologist? How was the selected tumour obtained? Just normal excision from the paraffin section? ïƒ AUTHOR RESPONSE: Thank you for the suggestion. We have now expanded these methods in Lines 132-138.
-Section 2.3. cat ref# of all these kits? ïƒ AUTHOR RESPONSE: Thank you for the suggestion. We have now added the cat ref#
-Section 2.4. cat ref# of all these kits? ïƒ AUTHOR RESPONSE: Thank you for the suggestion. We have now added the cat ref#
-Section 2.4. cat ref# of MiSeq Reagent kit v2 (300 cycles) all these kits? ïƒ AUTHOR RESPONSE: Thank you for the suggestion. We have now added the cat ref#
-Section 2.4. version of VariantStudio software (Illumina, Inc.) ïƒ AUTHOR RESPONSE: Thank you for the suggestion. We have now added the version.
-Section 2.5. why position R132? Succintly describe the method ïƒ AUTHOR RESPONSE: Thank you for the suggestion. We have explained why position R132 was analyzed (it is only mutant locus described for IDH1 in gliomas) and we have also briefly described the methods for BEAMing analysis in lines 161-199.
-Section 2.7. Ethics number? ïƒ AUTHOR RESPONSE: Thank you for the suggestion. We have added the Ethics number in this section (it is also present at the end of the manuscript in the Ethics Statement, as per de journal format).
7) Results
-Table 1: Looks like there is a formatting issue DATA NGS/IHC is a bit confusing to read! ïƒ AUTHOR RESPONSE: Thank you for the suggestion. We have reformatted the table to make it easier to read.
-Table 2: We have no colours in this table?! Table should include some padding spaces to improve readability ïƒ AUTHOR RESPONSE: Thank you for the suggestion. We have added the blue and red colors and included some separations to make it more easily readable.
Figure 1: Had to zoom in over 10x to be able to see anything and still could not observe the CT scan details and cytometry graphs lose pixel quality as well. The office should make sure that this image is provided with enough quality and make available high-quality file for download/visualization! ïƒ AUTHOR RESPONSE: Thank you for the suggestion. We have splitted Figure 1 into 4 different figures. We have separated the MRI images for each patient into an individual figure in order to magnify it and make it much more easier to visualize. We have added a fourth figure including only the BEAMing figures from the three patients, making it much more easier to visualize.
Figure 3 legend: …but progressive disease “slowly occurred”…. meaning? ïƒ AUTHOR RESPONSE: Thank you for the suggestion. We have erased “slowly” in order to avoid being redundant since the figure is selfexplanatory and because these tumors progress at a slower rate than higher grade tumor because it was a grade II oligodendroglioma.
8) Discussion
-Lines 611-616: Why? What obstacle(s) did you have to overcome? ïƒ This is an important issue. Thank you for the suggestion! We have added a brief paragraph explaining why liquid biopsy has been more cumbersome to develop in gliomas than in other solid tumors in lines Lines 611-616.
-Lines 643-650: Doesn’t this high % of false-positives inviabilize the use of this technology in these cases? Not to mention financially wise ïƒ AUTHOR RESPONSE: This is an important issue. Thank you for the suggestion! We have added a paragraph explaining these issues in Lines 738-745.
-Ins’t it normal for tumours with IDH1 mutation to also show mutations in the IDH2 gene? Would you say that your findings suits previous literature on IDH1 and 2 gliomas? ïƒ AUTHOR RESPONSE: This is an important issue. Thank you for the suggestion! No, the coexistence of two different IDH mutations in a glioma is extremely rare with only 4 cases previously reported in patients with gliomas. However, the co-existence of two different IDH mutations has been described in patients with IDH-mutant acute myeloid leukemia, most often after treatment with IDH inhibitors. Therefore, this is an important finding from our study. We have added a few lines clarifying all this in Lines 815-826 and in Lines 829-834.
-Lines 708-711: what about comparing to NGS of the tumour’s biopsy? ïƒ Thank you for the suggestion. We have added some comments regarding the interest of comparing BEAMing with NGS for tumour profiling in lines 842-845.

Reviewer 3 Report
In this report the authors demonstrate a method to detect IDH1 mutations in circulating tumors. The status of IDH1 in gliomas is a powerful diagnostic tool because cancers harboring IDH1 mutations have a higher prognosis. Although the study is somewhat limited, it outlines an important method of detecting these mutations in circulating tumors. However, the authors do not put their study in the context of current glioma genomic studies. Additionally, the authors should discuss the heterozygous status of these IDH1 mutations especially considering that they identified a co-occurring mutation.
Major comments
1. The IDH1 R132H mutation must be heterozygous to produce the 2-hydroxyglutarate byproduct (see Pappula et al 2021 and Harman et al 2009). This heterozygous mutation is the reason for better prognosis. This should be discussed in the introduction. Furthermore, at lines 144 and 171 the authors identify one patient with an R132H and R132C mutations. Is this patient a homozygous mutant (e.g. with two alleles with different mutations)? Or are these mutations not clonal (e.g. in two different cells from the same tumor)? This should be discussed because it is an important characteristic of the function of IDH1. Additionally, the authors should also introduce and discuss IDH1 as a driver gene.
2. IDH1 has been recently characterized as a driver gene. The 2-hydroxyglutarate byproduct is a DNA damage agent and therefore this mutation causes a mutator phenotype (e.g. subsequent mutations accumulate). Additionally, other co-occurring mutations have been demonstrated in IDH1 mutant and IDH1 wild type (see Pappula et al). Although this is a limited study, the author should definitely discuss the possibility of this technique in investigating these co-occurring mutations for better diagnosis.
Minor comments
1. Please define ctDNA the first time it is used (e.g. circulating tumor DNA).
2. Line 130: “5 patients being IDH1-R132H wild type” If the are wild type how are they IDH1-R132H?
3. Please introduce IDH1 by spelling out the whole name (e.g. isocitrate dehydrogenase).
Author Response
REVIEWER 3:
In this report the authors demonstrate a method to detect IDH1 mutations in circulating tumors. The status of IDH1 in gliomas is a powerful diagnostic tool because cancers harboring IDH1 mutations have a higher prognosis. Although the study is somewhat limited, it outlines an important method of detecting these mutations in circulating tumors. However, the authors do not put their study in the context of current glioma genomic studies. Additionally, the authors should discuss the heterozygous status of these IDH1 mutations especially considering that they identified a co-occurring mutation. AUTHOR RESPONSE: We thank the reviewer for his kind words and for the time dedicated to reviewing our manuscript which have allowed to improve and enrich our manuscript. Please see our responses below:
Major comments
- The IDH1 R132H mutation must be heterozygous to produce the 2-hydroxyglutarate byproduct (see Pappula et al 2021 and Hartman et al 2009). This heterozygous mutation is the reason for better prognosis. This should be discussed in the introduction. Furthermore, at lines 144 and 171 the authors identify one patient with an R132H and R132C mutations. Is this patient a homozygous mutant (e.g. with two alleles with different mutations)? Or are these mutations not clonal (e.g. in two different cells from the same tumor)? This should be discussed because it is an important characteristic of the function of IDH1. Additionally, the authors should also introduce and discuss IDH1 as a driver gene. ïƒ AUTHOR RESPONSE: This is an important issue. Thank you for the suggestion! We have now discussed all these, as the reviewer suggests in the Introduction in Lines 61-75 and in the Discussion lines 815-826. We have also referenced the interesting manuscript by Pappula et al.
- IDH1 has been recently characterized as a driver gene. The 2-hydroxyglutarate byproduct is a DNA damage agent and therefore this mutation causes a mutator phenotype (e.g. subsequent mutations accumulate). Additionally, other co-occurring mutations have been demonstrated in IDH1 mutant and IDH1 wild type (see Pappula et al). Although this is a limited study, the author should definitely discuss the possibility of this technique in investigating these co-occurring mutations for better diagnosis. ïƒ AUTHOR RESPONSE: This is an important issue. Thank you for the suggestion! We have now discussed all these, as the reviewer suggests in the Introduction in Lines 61-75 and in the Discussion Lines 842-855.
Minor comments
- Please define ctDNA the first time it is used (e.g. circulating tumor DNA). ïƒ AUTHOR RESPONSE: Thank you for the suggestion. We have now corrected this as the author suggests.
- Line 130: “5 patients being IDH1-R132H wild type” If the are wild type how are they IDH1-R132H? ïƒ AUTHOR RESPONSE: Thank you for the suggestion. We have now corrected this as the author suggests.
- Please introduce IDH1 by spelling out the whole name (e.g. isocitrate dehydrogenase). ïƒ AUTHOR RESPONSE: Thank you for the suggestion. We have now corrected this as the author suggests.

Round 2
Reviewer 2 Report
The authors replied to all the questions and "sorted out" all the concerns I expressed before. The manuscript is now ready for publication.
Reviewer 3 Report
The authors have made significant changes to this version. This reviewer is satisfied.